# Plasma Campesterol Is Positively Associated with Carotid Plaques in Asymptomatic Subjects

**DOI:** 10.3390/ijms231911997

**Published:** 2022-10-09

**Authors:** Valéria Sutti Nunes, Edite Vieira Silva de Campos, Jamal Baracat, Victor França, Érica Ivana Lázaro Gomes, Raissa Peres Coelho, Edna Regina Nakandakare, Vanessa Helena Souza Zago, Eliana Cotta de Faria, Eder Carlos Rocha Quintão

**Affiliations:** 1Laboratorio de Lipides (LIM10), Hospital das Clinicas HCFMUSP, Faculdade de Medicina, Universidade de Sao Paulo, Sao Paulo 01246-900, SP, Brazil; 2Lipid Laboratory and Center for Medicine and Experimental Surgery, Department of Clinical Pathology, Faculty of Medical Sciences, State University of Campinas (UNICAMP), Campinas 13083-887, SP, Brazil; 3Faculty of Medical Sciences, Department of Radiology, Hospital of Clinics, State University of Campinas (UNICAMP), Campinas 13083-888, SP, Brazil; 4Centro de Ciências da Vida, Pontifical Catholic University of Campinas, Campinas 13034-68, SP, Brazil

**Keywords:** carotid atherosclerosis, intestinal absorption, carotid intima-media thickness, cholesterol synthesis, phytosterols

## Abstract

Background: Increased cholesterol absorption and reduced synthesis are processes that have been associated with cardiovascular disease risk in a controversial way. However, most of the studies involving markers of cholesterol synthesis and absorption include conditions, such as obesity, diabetes, dyslipidemia, which can be confounding factors. The present study aimed at investigating the relationships of plasma cholesterol synthesis and absorption markers with cardiovascular disease (CVD) risk factors, cIMT (carotid intima-media thickness), and the presence of carotid plaques in asymptomatic subjects. Methods: A cross-sectional study was carried out in 270 asymptomatic individuals and anthropometrical parameters, fasting plasma lipids, glucometabolic profiles, high-sensitivity C-reactive protein (hs-CRP), markers of cholesterol synthesis (desmosterol and lathosterol), absorption (campesterol and sitosterol), cIMT, and the presence of atherosclerotic plaques were analyzed. Results: Among the selected subjects aged between 19 and 75 years, 51% were females. Age, body mass index, systolic and diastolic blood pressure, total cholesterol, non-HDL-C, triglycerides, glucose, and lathosterol/sitosterol ratios correlated positively with cIMT (*p* ≤ 0.05). Atherosclerotic plaques were present in 19% of the subjects. A direct association of carotid plaques with campesterol, OR = 1.71 (95% CI = 1.04–2.82, *p* ≤ 0.05) and inverse associations with both ratios lathosterol/campesterol, OR = 0.29 (CI = 0.11–0.80, *p* ≤ 0.05) and lathosterol/sitosterol, OR = 0.45 (CI = 0.22–0.95, *p* ≤ 0.05) were observed in univariate logistic regression analysis. Conclusions: The findings suggested that campesterol may be associated with atherosclerotic plaques and the lathosterol/campesterol or sitosterol ratios suggested an inverse association. Furthermore, synthesis and absorption of cholesterol are inverse processes, and the absorption marker, campesterol, may reflect changes in body cholesterol homeostasis with atherogenic potential.

## 1. Introduction

The dynamics of body cholesterol homeostasis involve several pathways and play a key role in atherogenesis. Unbalances between cholesterol synthesis and its intestinal absorption may be associated with the progression of atherosclerotic disease [1,2]. This can be illustrated by the plasma levels of cholesterol synthesis and absorption markers [3,4], since their reciprocal relationship strengthens the data on the regulation of serum cholesterol concentrations, and suggests that not only total serum cholesterol levels but also differences in cholesterol homeostasis, may be associated with subclinical atherosclerosis [5,6].

A recent meta-analysis provided an overview of the levels of cholesterol absorption and synthesis markers in different metabolic disorders, such as obesity, type 2 diabetes mellitus (DM2), metabolic syndrome, and familial hypercholesterolemia (HF). Interestingly, the study reinforces the reciprocal patterns of these markers and highlights that these metabolic disorders are characterized by higher cholesterol synthesis or higher cholesterol absorption that could contribute in different ways to atherosclerosis development [2].

Observational and clinical intervention studies established the measurement of carotid intima-media thickness (cIMT) as an important and non-invasive marker for subclinical and asymptomatic atherosclerotic cardiovascular disease. Indeed, cIMT is strongly associated with classical cardiovascular risk factors, such as age, gender, race, hypercholesterolemia, hyperglycemia, and smoking, among other factors [7,8].

An advanced stage of atherosclerosis comprises the formation of atherosclerotic plaques, which is considered to be mainly a process of the intimal arterial layer. Like cIMT, the carotid plaque has also been associated with cardiovascular risk, which could be used as a prognostic marker for future cardiovascular events, however, the extent to which the increase in cIMT is related to the future development of the carotid plaque is not entirely clear [9,10]. The relationship between subclinical atherosclerosis determined by cIMT and changes in cholesterol homeostasis was previously explored, but the findings are still controversial [11,12,13].

Clinical trials of primary and secondary cardiovascular prevention have demonstrated that inhibition of cholesterol synthesis with lipid-lowering treatments reduces LDL-C in dyslipidemia patients with high cardiovascular risk [14]. However, a higher concentration of absorption markers associated with decreased synthesis markers in non-diabetic individuals was independently associated with established cardiovascular disease or carotid stenosis [15,16].

Given the above, it is necessary to define the contribution of factors that directly interfere with cholesterol metabolism [16,17] and to evaluate whether the serum markers of cholesterol synthesis and absorption are associated with carotid atherosclerosis in a population of asymptomatic healthy individuals.

## 2. Results

The baseline characteristics of the participants are shown in Table 1. The studied population comprised both sexes (51% female), with median age equal to 44 years old, normolipidemic, and non-obese (BMI 23 kg/m^2^). Regarding cIMT, 12% (*n* = 34) present values above the cut-off level ≥ 0.9 mm. The presence of carotid plaques was reported for 19% (*n* = 51) of the study group.

Spearman’s correlation test between mean cIMT, clinical and biochemical parameters related to CVD risk and cholesterol synthesis and absorption markers are shown in Table 2. Spearman’s correlations analysis found positives correlations, as expected, between cIMT and the following variables: age (r = 0.66—*p* < 0.001), BMI (r = 0.16—*p* = 0.01), SBP (r = 0.34—*p* < 0.001), DBP (r = 0.19—*p* = 0.001), glucose (r = 0.25—*p* < 0.001), TC (r = 0.34—*p* < 0.001), non-HDL-C (r = 0.37—*p* < 0.001), triglycerides (r = 0.20—*p* = 0.001), LDL-C (r = 0.36—*p* < 0.001), VLDL-C (r = 0.20—*p* = 0.001), and hs-CRP (r = 0.17—*p* = 0.006). There were no correlations of cIMT with cholesterol synthesis (lathosterol, desmosterol) and absorption markers (campesterol, sitosterol), but a positive, however weak, correlation was found for the lathosterol/sitosterol ratio (r = 0.13—*p* = 0.041).

Relationships of cholesterol synthesis and absorption markers with classical CVD risk factors, using univariate and multivariate linear regression analyses are presented in Table 3. For the synthesis markers, the desmosterol was directly associated with triglycerides, but with hs-CRP the desmosterol association was inversely related. In the multiple analysis hs-CRP was inversely associated with desmosterol, a relationship explained in 3%. Lathosterol was directly associated with BMI, triglycerides and higher values were found in males; in the multiple analysis there was a direct association with BMI, a relationship explained in 4%.

Campesterol was inversely associated with age, BMI, triglycerides, and hs-CRP. In the multivariate analysis the set that best explains plasma campesterol concentrations were age, and hs-CRP inversely, explaining at 9%. Sitosterol was inversely associated with age, BMI, triglycerides, and hs-PCR; in the multivariate analysis, the group that best explains the plasma concentrations of sitosterol was inversely associated with BMI and hs-CRP, explaining 6%.

The desmosterol/campesterol ratio was directly associated with BMI and triglycerides; the variable that best relates with age in 2%. Desmosterol/sitosterol was directly associated with BMI and triglycerides. No variables were selected for multivariate analysis. Lathosterol/campesterol was directly associated with BMI, triglycerides and hs-CRP. The variables that best relate directly are BMI and hs-CRP in 8%. Lathosterol/sitosterol was directly associated with BMI and triglycerides; in multivariate, the relationship is direct with BMI in 9%.

The percentages of these relationships in the multiple analysis were low because the variables presented low relations. The univariate and multivariate linear regression analyses were performed to investigate the associations between cIMT and the cholesterol synthesis and absorption markers. The analysis was adjusted for age, sex, BMI, total cholesterol and non-HDL, and no significant associations were found.

The subjects were stratified according to the presence or absence of atherosclerotic plaques. The comparison data are presented according to their clinical, anthropometrical, and biochemical characteristics, as shown in Table 4. Among the 270 individuals who underwent carotid ultrasound, 19% had atherosclerotic plaques (N = 51), and the median age was 55 years. There was no difference between sexes in the groups. Mean cIMT among those with plaque was 0.9 mm (0.7–1.2 mm), and the mean cIMT was 39% higher in subjects with plaque.

Age, blood pressure, atherogenic lipoproteins, glucose, and hs-CRP presented significantly higher values in the group with plaques, but the alterations presented were among the reference values cited in the literature. There was a 20% reduction in the lathosterol/campesterol ratio. Such proportion provides an overall assessment of cholesterol homeostasis, as they consider the relative contributions of cholesterol synthesis and absorption [18].

Univariate logistic regression analysis for the presence of plaques and cholesterol synthesis and absorption markers are shown in Table 5. Significant associations were found in the univariate logistic regression analyses, but none in the multivariate analyses. There was an increase of 71% (OR 1.71), *p* = 0.04, in the chance of plaques occurring for the increase of 1 unit of campesterol, and a reduction of 71% (OR = 0.29), *p* = 0.02, in the chance of plaques occurring with the increase of 1 unit of the lathosterol/campesterol ratio. There was a 55% reduction (OR 0.45), *p* = 0.04 in the chance of plaques occurring to increase one unit of the lathosterol/sitosterol ratio.

## 3. Discussion

This work showed that the increase in plasma concentrations of campesterol was directly and individually associated with the odds ratio for the occurrence of atherosclerotic plaques and with changes in cholesterol homeostasis, explained by the reasons for synthesis/absorption of cholesterol (lathosterol/campesterol and lathosterol/sitosterol) that were inversely associated with the odds ratio for the occurrence of atherosclerotic plaques. This suggests that cholesterol homeostasis imbalances are related to subclinical atherosclerosis and subsequent cardiovascular disease in asymptomatic individuals but not with the measure of cIMT [19].

We did not find associations between cIMT and the synthesis (desmosterol and lathosterol) and absorption (campesterol and sitosterol) cholesterol markers. Similarly, in a population of young Finnish men in their usual diet were not observed associations or correlations between cholesterol synthesis and absorption with a vascular structure measured by cIMT [12]. Our results corroborate those of Nunes et al. showing increased cholesterol synthesis and absorption characterized, respectively, as high concentrations of desmosterol and campesterol when properly corrected for age, BMI, WC, hypertension, TC, smoking, and HOMA-IR representing primary causes of CAD, but not of the common carotid artery atherosclerosis [13].

Of the individuals studied, 12% had carotid thickening and 19% had a carotid plaque. A meta-analysis published in 2019, gathered information from world populations and estimated a frequency of 28% for cIMT and 21% for carotid plaques, however, this frequency was estimated in populations that included risk factors, such as hypertension, diabetes, and old age [20].

When we evaluated other studies with individuals at high cardiovascular risk, that is, with a diagnosis of established CVD and carotid stenosis associated with cholesterol homeostasis, we observed high concentrations of absorption markers and decreased synthesis, suggesting that absorption markers can be highly independent predictors of significant CVD [6]. Another study involving a population of 583 hospital employees aged 25–60 years with no prevalent cardiovascular disease, found a correlation between cholesterol synthesis markers with cIMT, while cholesterol absorption markers showed a weakly negative correlation [5]. When assessing classic risk factors with cIMT in asymptomatic individuals, we found a positive association with age, sex, and blood pressure. However, these results have often been observed in studies involving metabolic syndrome, hypertension, diabetes, or cardiovascular disease [6,21].

In the present study, elevated levels of cholesterol synthesis markers (lathosterol) and ratios (lathosterol/campesterol and lathosterol/sitosterol) were positively associated with BMI. It has been observed in other studies with overweight or obese individuals, that increased cholesterol synthesis marker (lathosterol) is associated with increased resistance to insulin, increased inflammatory activity, and, consequently, obesity is considered indicative of high cardiovascular risk. However, these associations are reversed after diet-induced weight loss [2,22,23]. Cholesterol metabolism in patients with metabolic syndrome, insulin resistance, abdominal obesity, and diabetes generally exhibits reduced absorption and high cholesterol synthesis patterns [24].

The cholesterol absorption and synthesis markers showed expected inverse associations, reflecting an apparent compensatory change in cholesterol metabolism because synthesis markers for absorption (lathosterol/campesterol and lathosterol/sitosterol) reflect the hemostasis of body cholesterol [19,25].

Individuals without atherosclerotic plaque showed a higher synthesis/absorption ratio (lathosterol/campesterol), which could explain a compensatory mechanism of cholesterol body homeostasis. Additionally, there is an inverse relationship between endogenous synthesis and the intake of dietary cholesterol, since the change in one results in a compensatory and opposite change in the other, keeping the cholesterol content in the entire body regulated [24,25].

It is well known that dietary supplementation with plant sterol is associated with clinically relevant reductions in LDL-C concentrations, and its use in foods is approved as a therapeutic option to control hypercholesterolemia [26], however, it has been suggested that the use of long-term phytosterols to provide cardiovascular protection remains a knowledge gap [14,27,28,29].

The present work had some limitations. Being a cross-sectional study, the cardiovascular risk of individuals in the long term was not evaluated. Such limitations could be solved by a prospective observational study of the research participants, considering the eating habits that were not controlled before enrollment in the study. Therefore, we cannot exclude the possibility that dietary variations have substantially influenced the amount of plant sterols ingested.

Hence, considering the results presented, it is suggested that cholesterol synthesis and absorption are interrelated and play an important role in regulating homeostasis, and the cholesterol absorption marker, campesterol, may reflect changes in body cholesterol homeostasis with atherogenic potential.

## 4. Materials and Methods

### 4.1. Research Design

A cross-sectional study was carried out on 270 volunteers from both genders selected through a large sample of individuals who sought government primary care centers (*n* = 598.288) and assessed their lipid profiles; those who did not meet our study criteria were excluded.

The study was approved by the local institutional ethical committee of the State University of Campinas (registration number 2.617.807/2018). All participants provided informed written consent. All methods were by the approved guidelines and in agreement with the Ethical Principles for Medical Research Involving Human Subjects as stated by the Declaration of Helsinki. A detailed description was already described [30].

### 4.2. Clinical Protocol

In the admission (phase I), the anthropometric assessment was carried out through measurements of weight, height, and body mass index (BMI, kg/m^2^). Systolic and diastolic blood pressure (SBP and DBP, respectively) was determined by measurements of the brachial artery, maintaining the same predetermined position and time procedures. The race was self-reported and answered according to the volunteer’s will and criteria.

### 4.3. Biochemical Analysis

Blood samples were collected after 12 h fasting and serum and plasma were centrifugated (4 °C, 1000× *g*, 10 min), and stored at −80 °C until analysis. Samples were analyzed in the Modular Analytics EVO P (Roche, Basel, Switzerland) using Roche (Mannheim, Germany) reagents for total cholesterol (TC), HDL-C (high-density lipoprotein cholesterol) and triglycerides (TG), uric acid (enzymatic methods), urea, alanine aminotransferase (ALT), aspartate aminotransferase (AST), gamma-glutamyl-transferase (GGT), alkaline phosphatase (AP), and creatinine (kinetic methods).

LDL-C was calculated by the Friedewald equation, and VLDL-C (very low-density lipoprotein cholesterol) was estimated by the formula TG/5. Non-HDL-C also serves as a parameter for the assessment of dyslipidemia [31].

Apolipoproteins A-I, B-100, and lipoprotein (a) Lp(a) were quantified by nephelometry in an automated system (BNII/Marburg, Germany) using commercial reagents Dade-Behring^®^ (Manheinn, Germany).

The high-sensitivity C-reactive protein (hsPCR) was assessed by immunoturbidimetry using the Tina-quant^®^ CRP (Roche Diagnostics^®^, Mannheim, Germany). Insulin was determined using the Human Insulin ELISA commercial assay (Millipore, Billerica, MA, USA), and glucose, using Roche reagents (Mannheim, Germany). In addition, the Homeostasis Model Assessment (HOMA) index was determined using Calculator version 2.2.2 (University of Oxford, Oxford, UK) [32].

Intestinal cholesterol absorption (campesterol and sitosterol) and synthesis markers (desmosterol and lathosterol) were measured by gas chromatography coupled to a mass spectrophotometer (Shimadzu GCMS-QP 2010 Plus, Kyoto, Japan) with the GC/MS software solution version 2.5 [33,34,35,36]. Concentrations of plasma steroid markers were presented as relative values, correcting the total plasma cholesterol concentration [33]. The analysis was performed with selected ion monitoring (SIM) using one target and two qualifier ions. The selected ions, where the first number is the target ion and all others are qualifiers ions, utilized for sterol identification were: 217, 149, and 109 *m*/*z* for 5α-cholestane; 119, 351, and 253 *m*/*z* for desmosterol, 255, 213, and 458 *m*/*z* for lathosterol; and 129, 382, and 343 *m*/*z* for campesterol, and 129, 486, and 357 *m*/*z* for sitosterol. Quantitation by GC–MS/SIM was based on the peak area ratios of the target ions of the analyte to that of the internal standard (5α-cholestane) and compared to concentrations of matched calibration standards [34,35,36].

### 4.4. cIMT Measurements and Atherosclerotic Plaque Detection

The volunteers who attend the inclusion criteria described above were invited to cIMT measurements and atherosclerotic plaque detection, whose determinations were performed by a single-trained radiologist, blind to the study subjects. Determinations were performed by high-resolution B-mode ultrasonography using a 6–9 MHz linear array ultrasound imaging system ATL HDI 3500 Ultrasound System (Advanced Technology Laboratories Ultrasound, Bothell, WA, USA). All participants were examined in dorsal decubitus, with the head elevated at about 20° and rotated at 45°, and were performed five longitudinal measurements of segments of the left and right common carotid arteries at the distal wall and 1 cm from the bifurcation. The imaging was performed initially evaluating the common carotid artery with Antero-oblique insonation above the clavicle and along with the bulb and internal carotid artery, as previously described [37,38,39].

The values of cIMT above the 75th percentile are considered to be high and indicative of an increased risk of CVD [38]. As the study participants are asymptomatic individuals, we chose to use the 90th percentile; the reference value of the cIMT measure adopted was <0.90 mm, and values cIMT equal to or greater than 0.90 mm were considered carotid thickening [37,38].

In addition to cIMT, a scan investigation of the extracranial carotid of both sides of the plaques was performed. It was considered atherosclerotic plaque if the following conditions are observed: the focal wall thickening with at least 0.5 mm toward the vessel lumen, focal wall thickening that is at least 50% greater than that of the surrounding vessel wall, or focal region with cIMT above 1.5 mm that protrudes into the lumen that is distinct from the adjacent boundary [37,39].

### 4.5. Statistical Analyses

The statistical analyses were performed using SAS System for Windows 9.1.3 (Statistical Analysis System/SAS Institute Inc., 2002–2003, Cary, NC, USA) and SPSS 22.0. (IBM SPSS Statistics for Windows, Version 22.0. Armonk, NY, USA: IBM Corp).

The distribution of all parameters was tested with the Shapiro–Wilk normality test. Due to the absence of normality, the variables were expressed using median and interquartile ranges (25th and 75th percentiles). Numeric variables were transformed in ranks in the absence of normal distribution.

Correlations were made between cIMT with the classic CVD risk parameters and with cholesterol synthesis and absorption markers, as well as between cholesterol synthesis and absorption markers with clinical and biochemical parameters using Spearman’s correlation coefficients (rho/r), nonparametric test.

The total group was separated according to the presence or absence of atherosclerotic plaques and compared using the Wilcoxon—Mann–Whitney nonparametric test; the variable’s responses were transformed into ranks and adjusted for age, sex, BMI, total cholesterol, and non-HDL-C.

Univariate and multiple linear regression analysis was used to evaluate the associations between cIMT with classical CVD risk parameters and with cholesterol synthesis and absorption markers, where the models were adjusted for sex, age, BMI, total cholesterol, and non-HDL-C. The cholesterol synthesis and absorption markers were associated with anthropometrical and biochemical parameters. Results of multiple linear regression analysis are expressed as coefficients of determination (partial and total R2). The stepwise criterion method was used through multiple analyses.

Logistic regression analyzes were used to assess the relative risk through the calculation of the odds ratio (OR) between carotid atherosclerotic plaques and the other variables. Analyses were adjusted for age, sex, BMI, total cholesterol, and non-HDL-C. The tests were considered significant at a probability value (*p*) of 0.05 or less.

## 5. Conclusions

This study suggested a direct association of campesterol with carotid atherosclerotic plaques and an inverse association with rates of synthesis/absorption, thus reflecting changes in body cholesterol balance, and that increased absorption and decreased synthesis are associated with carotid atherosclerosis.

Although the present investigation is limited in that it is not prospective, it nevertheless has the advantage of excluding conditions that influence sterol markers of cholesterol metabolism, such as obesity, diabetes, and dyslipidemias, which are typically confounding factors in atherosclerosis.

## Figures and Tables

**Table 1 ijms-23-11997-t001:** Clinical, anthropometrical, and biochemical characteristics of all subjects.

Parameters	N	Medians (P25–P75)
Age (years)	270	44 (33–54)
Sex (F/M)	139/131	
Race (White/non-White)	226/44	
Plaques (without/with)	219/51	
BMI (kg/m^2^)	270	23 (21–25)
SBP (mmHg)	270	120 (110–126)
DBP (mmHg)	270	80 (70–80)
Mean cIMT (mm)	270	0.60 (0.50–0.72)
Glucose (mg/dL)	270	83 (78–88)
Insulin (uU/mL)	270	4 (2.4–5.8)
HOMA-IR	270	0.82 (0.46–1.20)
TC (mg/dL)	270	169 (147–194)
HDL-C (mg/dL)	270	58 (37–75)
Non-HDL-C (mg/dL)	270	114 (94–132)
LDL-C (mg/dL)	270	101 (85–117)
TG (mg/dL)	270	73 (55–96)
VLDL-C (mg/dL)	270	15 (11–19)
hs-CRP (mg/L)	270	0.8 (0.4–1.6)
Cholesterol Synthesis Markers *		
Desmosterol	253	0.32 (0.24–0.41)
Lathosterol	255	0.49 (0.35–0.72)
Cholesterol Absorption Markers *		
Campesterol	253	1.18 (0.85–1.62)
Sitosterol	251	1.27 (0.74–2.06)
Synthesis/absorption ratios *		
Desmosterol/Campesterol	253	0.26 (0.16–0.41)
Desmosterol/Sitosterol	251	0.21 (0.14–0.43)
Lathosterol/Campesterol	253	0.41 (0.26–0.68)
Lathosterol/Sitosterol	251	0.38 (0.22–0.77)

N = absolute number: Data are represented as median (interquartile range: 25th percentile 75th percentile); F = Female; M = Male; BMI = Body mass index; SBP = systolic blood pressure; DBP = diastolic blood pressure; cIMT = carotid intima-media thickness; HOMA = homeostasis model assessment; TC = total cholesterol; HDL-C = high-density lipoprotein; LDL-C = low-density lipoprotein; TG = triglycerides; VLDL-C = very low-density lipoprotein; hs-CRP = high-sensitivity C-reactive protein; * 102 ug/mL/cholesterol as mg/dL.

**Table 2 ijms-23-11997-t002:** Spearman’s correlation coefficients of cIMT with classical CVD risk parameters and with cholesterol synthesis and absorption markers.

Parameters	r	*p*-Values
Age (years)	0.66	0.000
BMI (kg/m^2^)	0.16	0.011
SBP (mmHg)	0.34	0.000
DBP (mmHg)	0.19	0.001
Glucose (mg/dL)	0.26	0.000
Insulin (uU/mL)	0.00	NS
HOMA-IR	0.06	NS
TC (mg/dL)	0.34	0.000
HDL-C (mg/dL)	0.09	NS
Non-HDL-C (mg/dL)	0.37	0.000
LDL-C (mg/dL)	0.36	0.000
TG (mg/dL)	0.20	0.001
VLDL-C (mg/dL)	0.20	0.001
hs-CRP (mg/L)	0.17	0.006
Cholesterol Synthesis Markers *		
Desmosterol	−0.01	NS
Lathosterol	0.09	NS
Cholesterol Absorption Markers *		
Campesterol	−0.10	NS
Sitosterol	−0.10	NS
Synthesis/absorption ratios *		
Desmosterol/Campesterol	0.06	NS
Desmosterol/Sitosterol	0.07	NS
Lathosterol/Campesterol	0.11	NS
Lathosterol/Sitosterol	0.13	0.041

Number = 238–270; r = Spearman’s correlation coefficient: *p* ≤ 0.05 significance; all variables transformed into ranks due of the absence of normal distribution. Number = 238–270; BMI = Body mass index; SBP = systolic blood pressure; DBP = diastolic blood pressure; TC = total cholesterol; HDL-C = high-density lipoprotein cholesterol; LDL-C = low-density lipoprotein cholesterol; TG = triglycerides; VLDL-C = very lowdensity lipoprotein cholesterol; hs-CRP = high-sensitivity C-reactive protein; NS = non-significance; * 10^2^ µg/mL/cholesterol as mg/dL.

**Table 3 ijms-23-11997-t003:** Relationships of markers of cholesterol synthesis and absorption with classical CVD risk factors.

Independent Variables	Univariate		Multivariate			
	Β	*p*-Values	β	*p*-Values	Partial R^2^	Total R^2^
Desmosterol (*n* = 241)						
Sex (F/M)	32.50 > M	0.001				0.03
TG (mg/dL)	0.47	0.004				
hs-CRP (mg/L)	−2.27	0.03	−4.29	0.004	0.03	
Lathosterol (*n* = 241)						
Sex (F/M)	24.18 > M	0.01				0.04
BMI (kg/m^2^)	8.95	<0.0001	5.88	0.002	0.04	
TG (mg/dL)	0.70	<0.0001				
Campesterol (*n* = 240)						
Age (years)	−0.85	0.019	−1.31	0.001	0.03	0.09
BMI (kg/m^2^)	−6.71	0.0003				
TG (mg/dL)	−0.74	<0.0001				
hs-CRP (mg/L)	−3.00	0.006	−5.15	0.01	0.06	
Sitosterol (*n* = 240)						
Age (years)	−0.71	0.05				0.06
BMI (kg/m^2^)	−7.57	<0.0001	−5.26	0.01	0.04	
TG (mg/dL)	−0.57	0.0003				
hs-CRP (mg/L)	−2.12	0.05	−3.22	0.02	0.02	
Desmosterol/Campesterol (*n* = 240)						
Age (years)			1.03	0.01	0.02	0.02
BMI (kg/m^2^)	4.10	0.03				
TG (mg/dL)	0.80	<0.0001				
Desmosterol/Sitosterol (*n* = 241)						
BMI (kg/m^2^)	5.32	0.004				
TG (mg/dL)	0.71	<0.0001				
Lathosterol/Campesterol (*n* = 240)						
BMI (kg/m^2^)	10.31	<0.0001	7.07	0.001	0.06	0.08
TG (mg/dL)	0.91	<0.0001				
hs-CRP (mg/L)			2.82	0.04	0.02	
Lathosterol/Sitosterol (*n* = 238)						
BMI (kg/m^2^)	10.55	<0.0001	7.6	0.0001	0.09	0.09
TG (mg/dL)	0.75	<0.0001				

*n* = number; β = estimated parameter; Linear Regression univariate and multivariate; *p* ≤ 0.05 significance; all variables transformed into ranks due of the absence of normal distribution; F = female; M = male; TG = triglycerides; BMI = Body mass index; hs-CRP = high-sensitive C-reactive protein.

**Table 4 ijms-23-11997-t004:** Comparative analyses of clinical, anthropometrical, and biochemical characteristics of subjects according to the presence or absence of atherosclerotic plaques.

Group	Without Plaques	With Plaques	*p* ^
Sex (*n*, F/M)	115/104	24/27	NS
Parameters	Medians (P25–P75)	Medians (P25–P75)	
Age (years)	41 (30–50)	55 (49–62)	0.0001
BMI (kg/m^2^)	23 (22–25)	23 (21–25)	NS
SBP (mmHg)	120(110–120)	130 (120–140)	0.0001
DBP (mmHg)	80(70–80)	80 (80–90)	0.001
Mean cIMT (mm)	0.6 (0.5–0.7)	0.9 (0.7–1.2)	0.0001
Glucose (mg/dL)	83 (78–88)	86 (81–91)	0.020
TC (mg/dL)	166 (144–192)	183 (167–206)	0.0001
HDL-C (mg/dL)	55 (37–75)	66 (39–74)	NS
Non-HDL-C (mg/dL)	111 (92–128)	128 (116–140)	0.0001
LDL-C (mg/dL)	99 (82–114)	113 (100–122)	0.0001
Triglycerides (mg/dL)	71 (53–91)	84 (65–102)	0.010
VLDL-C (mg/dL)	14 (11–18)	17 (13–20)	0.001
hs-CRP (mg/L)	0.8 (0.4–1.4)	1.2 (0.7–2.5)	0.002
Cholesterol Synthesis Markers *			
Desmosterol	0.32 (0.23–0.41)	0.36 (0.28–0.43)	NS
Lathosterol	0.50 (0.35–0.75)	0.48 (0.33–0.64)	NS
Cholesterol Absorption Markers *			
Campesterol	1.15 (0.83–1.58)	1.38 (0.93–1.76)	NS
Sitosterol	1.22 (0.73–2.01)	1.50 (0.89–2.17)	NS
Synthesis/absorption ratios *			
Desmosterol/Campesterol	0.26 (0.16–0.42)	0.26 (0.15–0.37)	NS
Desmosterol/Sitosterol	0.22 (0.14–0.44)	0.19 (0.15–0.42)	NS
Lathosterol/Campesterol	0.43 (0.28–0.75)	0.35 (0.23–0.49)	0.030
Lathosterol/Sitosterol	0.39 (0.22–0.81)	0.29 (0.21–0.55)	NS

Data are represented as median (interquartile range 25–75th percentiles); Number (*n*) of subjects: without plaques 203–219; with plaques 48–51; F = Female; M = Male; BMI = Body mass index; SBP = systolic blood pressure; DBP = diastolic blood pressure; cIMT = carotid intima-media thickness; TC = total cholesterol; HDL-C = high-density lipoprotein; LDL-C = low-density lipoprotein; VLDL-C = very low density lipoprotein; hs-CRP = high-sensitivity C-reactive; ^ Chi-square test (X2); Mann–Whitney Wilcoxon W test, *p*-values ≤ 0.05 significance; NS = non-significance; * 10^2^ µg/mL/cholesterol as mg/dL.

**Table 5 ijms-23-11997-t005:** Univariate logistic regression analyses for the presence of plaques.

Variables	Unit	OR	CI 95% (OR)	*p*-Values
Desmosterol *	1.0	5.39	(0.71–41.03)	NS
Lathosterol *	1.0	0.36	(0.10–1.28)	NS
Campesterol *	1.0	1.71	(1.04–2.82)	0.04
Sitosterol *	1.0	1.26	(0.93–1.71)	NS
Desmosterol/campesterol	1.0	0.46	(0.09–2.36)	NS
Desmosterol/Sitosterol	1.0	0.70	(0.21–2.35)	NS
Lathosterol/campesterol	1.0	0.29	(0.11–0.80)	0.02
Lathosterol/Sitosterol	1.0	0.45	(0.22–0.95)	0.04

OR = Odds Ratio; CI = confidence interval; *p* ≤ 0.05; NS = non-significance. Logistic Regression Analysis (univariate model) adjusted by: age, gender, BMI (Body mass index), TC and non-HDL; * 10^2^ ug/mL/cholesterol as mg/dL.

## Data Availability

The data presented in this study are available upon request from the corresponding author.

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
