# Peer review of "Plasma Campesterol Is Positively Associated with Carotid Plaques in Asymptomatic Subjects"

_ijms, 2022, doi:10.3390/ijms231911997_

Round 1

Reviewer 1 Report

excellent scientific path and quality of results for highlighted

Author Response

Answers provided to Reviewer 1:

In the paragraphs below we corrected punctuation errors but found no misspellings:

Lines127 – 131:

Among the 270 individuals who underwent carotid ultrasound, 19% had atherosclerotic plaques (N = 51), and the median age was 55 years. There was no difference between sexes in the groups. Mean cIMT among those with plaque was 0.9 mm (0.7-1.2 mm), and the mean cIMT was 39% higher in subjects with plaque.

Lines 167-171:

Of the individuals studied, 12% had carotid thickening and 19% of individuals had a carotid plaque. A meta-analysis published in 2019, gathered information from world populations and estimated a frequency of 28% for EMI and 21% for carotid plaques, however, this frequency was estimated in populations that included risk factors such as hypertension, diabetes, and old age [30].

Lines 172-182;

When we evaluated other studies with individuals at high cardiovascular risk, that is, with a diagnosis of established CVD and carotid stenosis associated with cholesterol homeostasis, we observed high concentrations of absorption markers and decreased synthesis, suggesting that absorption markers can be highly independent predictors of significant CVD  [6]. Another study involving a population of 583 hospital employees aged 25-60 years with no prevalent cardiovascular disease, found a correlation between cholesterol synthesis markers with cIMT, while cholesterol absorption markers showed a weakly negative correlation [5]. When assessing classic risk factors with cIMT in asymptomatic individuals, we found a positive association with age, sex, and blood pressure. However, these results have often been observed in studies involving metabolic syndrome, hypertension, diabetes, or cardiovascular disease [6,21].

Lines 183-190:

In the present study elevated levels of cholesterol synthesis markers (lathosterol) and ratios (lathosterol/campesterol and lathosterol/sitosterol) were positively associated with BMI. It has been observed in other studies with overweight or obese individuals, that increased cholesterol synthesis marker (lathosterol) is associated with increased resistance to insulin, increased inflammatory activity, and consequently, obesity is considered indicative of high cardiovascular risk. However, these associations are reversed after diet-induced weight loss [2,22,23].  Cholesterol metabolism in patients with metabolic syndrome, insulin resistance, abdominal obesity and diabetes generally exhibits reduced absorption and high cholesterol synthesis patterns [24].

Lines 242-244:

LDL-C was calculated by the Friedewald equation, and VLDL-C (very low-density lipoprotein cholesterol) was estimated by the formula TG/5. Non-HDL-C also serves as a parameter for the assessment of dyslipidemia [31].

Lines 260-266:

The selected ions, where the first number is the target ion and all others are qualifiers ions, utilized for sterol identification were: 217, 149, and 109m/z for 5α-cholestane; 119, 351, and 253m/z for desmosterol, 255, 213 and 458 m/z for lathosterol; and 129, 382 and 343m/z for campesterol, and 129, 486 and 357m/z for sitosterol. Quantitation by GC–MS/SIM was based on the peak area ratios of the target ions of the analyte to that of the internal standard (5α-cholestane) and compared to concentrations of matched calibration standards [34–36].

Lines 279-283:

The values of cIMT above the 75th percentile are considered to be high and indicative of an increased risk of CVD [38]. As the study participants are asymptomatic individuals, we chose to use the 90th percentile; the reference value of the cIMT measure adopted was <0.90 mm, and values cIMT equal to or greater than 0.90 mm were considered carotid thickening [37,38].

Lines 284-286:

In addition to cIMT, a scan investigation of the extracranial carotid of both sides of the plaques was performed. It was considered atherosclerotic plaque if the following conditions are observed:

Lines 302-305:

The total group was separated according to the presence or absence of atherosclerotic plaques and compared using the Wilcoxon - Mann Whitney nonparametric test; the variable’s responses were transformed into ranks and adjusted for age, sex, BMI, total cholesterol, and non-HDL-C.

Lines 306-316:

Univariate and multiple linear regression analysis was used to evaluate the associations between cIMT with classical CVD risk parameters and with cholesterol synthesis and absorption markers, where the models were adjusted for sex, age, BMI, total cholesterol, and non-HDL-C.  The cholesterol synthesis and absorption markers were associated with anthropometrical and biochemical parameters. Results of multiple linear regression analysis are expressed as coefficients of determination (partial and total R2). The stepwise criterion method was used through multiple analyses.

Logistic regression analyzes were used to assess the relative risk through the calculation of the odds ratio (OR) between carotid atherosclerotic plaques and the other variables. Analyses were adjusted for age, sex, BMI, total cholesterol, and non-HDL-C. The tests were considered significant at a probability value (p) of 0.05 or less.

Reviewer 2 Report

Plasma Campesterol is Positively Associated with Carotid Plaques in Asymptomatic Subjects by Nunes et al., describes the association of plasma campesterol (a biomarker for cholesterol absorption) is associated with carotid plaques in asymptomatic patients. I don't believe the title of the paper is a true statement of the data presented. While OR for campesterol was high in patients with plaques, when balanced with other non-cholesterol sterols the outcome OR reduces to less than 1 indicating  a balance between cholesterol absorbers/synthesizer concentrations observed. While I appreciate the authors have found an association of a non-cholesterol synthesizer (campesterol) to be associated with potential CVD risk, I don't believe the data shown definitively indicates that it can be used as a marker of atherogenic potential. 

There were no correlations of cIMT with cholesterol synthesis (lathosterol, desmosterol) and absorption markers (campesterol,sitosterol). The patients stratified based on the presence of atherosclerotic plaques also had no correlation with cholesterol synthesis and absorption markers either. Given these two major statements, can the authors conclude that camposterol has atherogenic potential?

Other issues I found while reviewing the paper are as follows;

While it may be inferred that this is a healthy population, there is no mention of the patients medication/health records. Are these patients generally healthy (apart from those stratified with the highest arterial plaques)? 

Data pertaining to Table 3 is written in a confusing manner (Line 99-119).

It appears that the measure of cholesterol synthesis/absorption markers are associated with parameters of CVD risk but the study was done over (what appears to be) one visit. Cholesterol absorption and synthesis processes are dynamic and the static accrual of the data may not truly reflect the power of non-cholesterol sterol concentrations in their utility as a biomarker. 

Author Response

Answers provided to Reviewer 2:

In some paragraphs, we corrected punctuation errors but found no misspellings.

The Reviewer stated “While I appreciate the authors have found an association of a non-cholesterol synthesizer (campesterol) to be associated with potential CVD risk, I don't believe the data shown definitively indicates that it can be used as a marker of atherogenic potential”.

Answer: The authors believe the Reviewer meant a non-cholesterol absorption marker (campesterol), not a synthesizer...

The authors stated:  ” This study suggested a direct association of campesterol with carotid atherosclerotic plaques and an inverse association with rates of synthesis/absorption, thus reflecting changes in body cholesterol balance, and that increased absorption and decreased synthesis are associated with carotid atherosclerosis”. 

Thus we agree with the Reviewer’s comment that present data cannot indicate campesterol is a marker of atherogenic potential:  “I don't believe the data shown definitively indicates that it can be used as a marker of atherogenic potential”.

The Reviewer stated: “Given these two major statements, can the authors conclude that campesterol has atherogenic potential?”

Answer: Campesterol concentration is a marker of cholesterol absorption. As such, campesterol may indicate that increased absorption and decreased synthesis are associated with carotid atherosclerosis.

Reviewer’scomment:  While it may be inferred that this is a healthy population, there is no mention of the patients medication/health records. Are these patients generally healthy (apart from those stratified with the highest arterial plaques)?

Answer: This is definitely a healthy population not treated with drugs that influence lipid metabolism.

Reviewer’s comment:  Data pertaining to Table 3 is written in a confusing manner (Line 99-119).

Answer: Table 3 was constructed to emphasize to the reader of this work that there are several conventional risk factors for atherosclerosis risk that have been systematically omitted in studies in which plasma sterols are analyzed, as is the case with this work. Hence the importance of demonstrating that such factors are present in our population, although they are related to atherosclerosis on a small scale. Omitting such information represents undesirable carelessness.

Reviewer’s comment: It appears that the measure of cholesterol synthesis/absorption markers are associated with parameters of CVD risk but the study was done over (what appears to be) one visit. Cholesterol absorption and synthesis processes are dynamic and the static accrual of the data may not truly reflect the power of non-cholesterol sterol concentrations in their utility as a biomarker. 

Answer: we ask the Editor to add the following paragraph to our conclusions on line 322:

Although the present investigation is limited in that it is not prospective, it

nevertheless has the advantage of excluding conditions that influence

sterol markers of cholesterol metabolism, such as obesity, diabetes, and

dyslipidemias which are typically confounding factors in atherosclerosis.

Round 2

Reviewer 2 Report

I appreciate the comments/rebuttal from the authors and feel the manuscript better represents the findings.